# TEMPERATURE AS A META-POLICY: ADAPTIVE TEMPERATURE IN LLM REINFORCEMENT LEARNING

**Haoran Dang**[1]   **Cuiling Lan**[2*]   **Hai Wan**[1]   **Xibin Zhao**[1*]   **Yan Lu**[2]

[1]Tsinghua University   [2]Microsoft Research Asia

danghr23@mails.tsinghua.edu.cn, {culan,yanlu}@microsoft.com,
{wanhai,zxb}@tsinghua.edu.cn

## ABSTRACT

Temperature is a crucial hyperparameter in large language models (LLMs), controlling the trade-off between exploration and exploitation during text generation. High temperatures encourage diverse but noisy outputs, while low temperatures produce focused outputs but may cause premature convergence. Yet static or heuristic temperature schedules fail to adapt to the dynamic demands of reinforcement learning (RL) throughout training, often limiting policy improvement. We propose Temperature Adaptive Meta Policy Optimization (TAMPO), a new framework that recasts temperature control as a learnable meta-policy. TAMPO operates through a hierarchical two-loop process. In the inner loop, the LLM policy is updated (*e.g.*, using GRPO) with trajectories sampled at the temperature selected by the meta-policy. In the outer loop, meta-policy updates the distribution over candidate temperatures by rewarding those that maximize the likelihood of high-advantage trajectories. This trajectory-guided, reward-driven mechanism enables online adaptation without additional rollouts, directly aligning exploration with policy improvement. On five mathematical reasoning benchmarks, TAMPO outperforms baselines using fixed or heuristic temperatures, establishing temperature as an effective learnable meta-policy for adaptive exploration in LLM reinforcement learning.

## 1 INTRODUCTION

Reinforcement learning (RL) has become a promising paradigm for aligning large language models (LLMs) with human preferences and task-specific objectives (Ziegler et al., 2019; Ouyang et al., 2022; Bai et al., 2022; Chen et al., 2025). Traditional RLHF approaches often rely on PPO-based RL-based post-training, which requires a learned value network and incurs significant computational overhead. Recent critic-free algorithms, such as GRPO (Shao et al., 2024; Guo et al., 2025) and REINFORCE++ (Hu et al., 2025), demonstrate that large-scale LLM reinforcement learning can be both scalable and stable, bypassing the need for value networks while maintaining performance.

One of central challenges in RL remains the exploration–exploitation trade-off (Sutton et al., 1998; Kaelbling et al., 1996). For LLMs, sampling temperature serves as a direct and interpretable control knob: higher temperature produces a more uniform (random) distribution, encouraging diverse but potentially noisy generations, while lower temperature concentrates probability mass, favoring precision but at the risk of missing promising alternatives. Existing approaches, however, treat temperature as fixed or manually tuned, ignoring feedback from the learning process. Popular critic-free RL algorithms (*e.g.*, GRPO) (Shao et al., 2024; Guo et al., 2025; Hu et al., 2025; Yu et al., 2025) generate multiple rollouts at a given temperature to estimate trajectory advantages and policy gradients, but never adapt temperature based on trajectory outcomes.

We argue that temperature should be treated as a decision variable, not a fixed hyperparameter for LLM RL. Unlike entropy regularization coefficients or KL penalties, which influence exploration (Guo et al., 2025; Shen, 2025), temperature directly modulates the sampling distribution over text outputs in a simple, transparent manner. This motivates our work on principled, trajectory-guided temperature adaptation for effective LLM policy learning.

---

*Corresponding author.

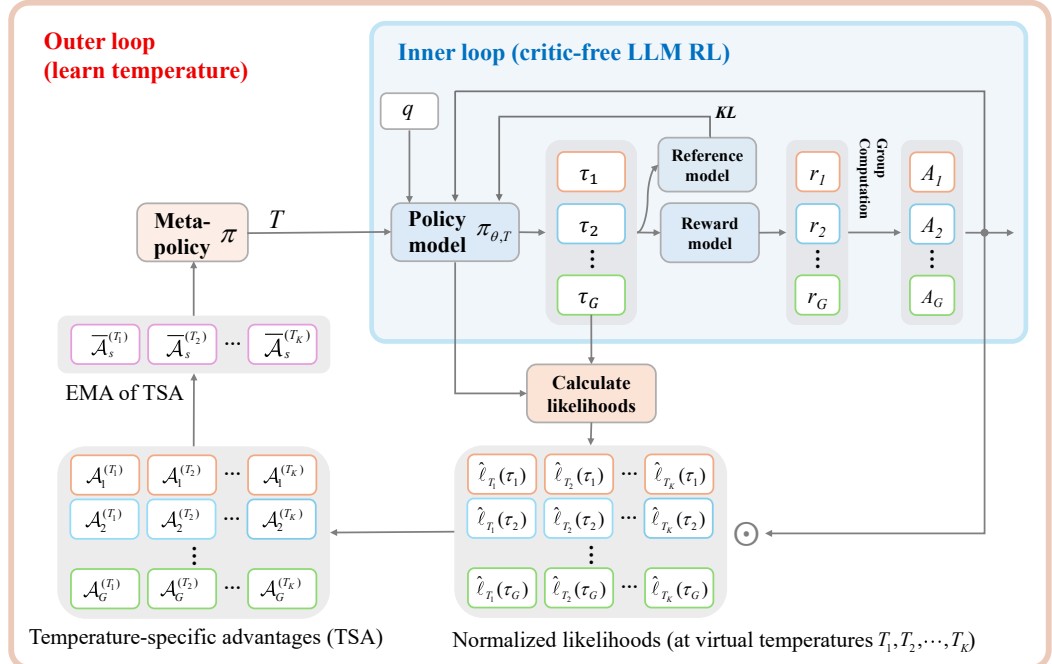

Figure 1: Overview of Temperature Adaptive Meta Policy Optimization (TAMPO). The framework operates through a hierarchical two-loop process. In the inner loop, the LLM policy is optimized with critic-free RL (*e.g.*, GRPO) using rollouts sampled at the temperature chosen by the meta-policy. In the outer loop, the meta-policy is updated by evaluating trajectory likelihoods under virtual temperatures, deriving temperature-specific advantages ($\mathcal{A}_i^{(T_k)} = \hat{\ell}_{T_k}(\tau_i) \cdot A_i$ for trajectory $\tau_i$ w.r.t. virtual temperature $T_k$), and reinforcing those that yield high-advantage rollouts (see §3). This design establishes temperature as a learnable meta-policy, enabling online adaptation and effective optimization of LLM policy without extra rollouts.

In this work, we introduce Temperature Adaptive Meta Policy Optimization (TAMPO), a meta-learning framework that jointly optimizes the LLM policy, and a meta-policy over temperatures. Figure 1 shows the overall framework, which operates through a hierarchical two-loop process. In the inner loop, the LLM policy model $\pi_{\theta,T}$ is optimized using critic-free RL (*e.g.*, GRPO) at a sampled temperature $T$. In the outer loop, a meta-policy $\pi$ is obtained based on temperature-specific advantages by reusing the inner loop rollouts, reinforcing temperatures that are more likely to generate trajectories with high advantages.

*Intuitively, the temperatures that facilitate discovering rewarding outputs are reinforced, while the ineffective ones are suppressed, allowing the learned temperature to dynamically align with policy improvement.*

We summarize our main contributions as follows:

- We formulate temperature as a learnable meta-policy in LLM RL, reframing temperature selection as a policy optimization problem to enhance adaptive, reward-driven exploration.

- We propose TAMPO, a hierarchical framework that jointly updates the LLM policy and a meta-policy over temperatures, rewarding those that prompt high-advantage rollouts.

- On five challenging mathematical reasoning benchmarks, TAMPO achieves better performance than baselines using fixed or heuristic temperatures, demonstrating the effectiveness of our trajectory-guided adaptive temperature control in LLM reinforcement learning.

TAMPO provides a principled, end-to-end, feedback-driven mechanism to dynamically balance exploration and exploitation, eliminating manual temperature sweeps, and improving the effectiveness of RL-based post-training.

## 2 RELATED WORK

**Critic-Free RL Methods.** Critic-free RL algorithms, such as REINFORCE Leave-One-Out (RLOO) (Kool et al., 2019), GRPO (Shao et al., 2024; Guo et al., 2025), DAPO (Yu et al., 2025), and REINFORCE++ (Hu et al., 2025) eliminate the need for learning value networks (critics), making them more efficient and scalable for LLM RL-based post-training. However, these approaches still mainly rely on fixed sampling temperatures, which may lead to under-exploration when the training temperature is too low, or wasted computation and noisy samples when too high. Our method complements these algorithms by learning a meta-policy over temperatures, which adapts sampling based on trajectory outcomes to better guide LLM policy optimization.

**Exploration–Exploitation in RL.** The exploration–exploitation dilemma is a fundamental challenge in RL, where agents must balance between exploring new actions to discover potentially better strategies and exploiting known actions to maximize immediate rewards (Wang et al., 2025). Traditional methods, such as $\epsilon$-greedy, temperature annealing, and upper confidence bounds (UCB), employ fixed or heuristic schedules to manage this balance (Wikipedia contributors, 2025)(Sutton et al., 1998). Typically, $\epsilon$-greedy linearly or exponentially decays $\epsilon$ over time, temperature annealing gradually lowers the sampling temperature to reduce exploration, and UCB adaptively selects actions based on upper confidence bounds that balance estimated value and uncertainty. In RL for LLMs, maintaining exploration is crucial to avoid premature convergence to suboptimal reasoning paths. Nucleus sampling (Holtzman et al., 2019) provides one practical strategy by restricting sampling to the smallest set of tokens whose cumulative probability exceeds a threshold $p$, thereby adaptively balancing diversity and reliability in generation, where $p$ is in general fixed (*e.g.*, 0.95). Entropy regularization is widely adopted to promote diverse outputs by encouraging higher-entropy policies (Guo et al., 2025; Shen, 2025). Sethi *et al.*(Sethi et al., 2025) view policy optimization as a continuous-time dynamical system and gradually decay the entropy regularization weight (typically as $1/t$). Shen (Shen, 2025) constrains entropy within a pre-defined range. However, the optimal entropy level during training remains unclear.

Beyond entropy, sampling temperature offers a direct and interpretable control knob for balancing exploration and exploitation in LLMs. Du *et al.* (Du et al., 2025) propose an adaptive inference-time method that selects optimal temperatures using multiple sampled generations. While effective for inference, it does not address how temperature can be dynamically optimized during RL training. Current LLM RL approaches either fix the temperature (Guo et al., 2025; Chen et al., 2025) or manually tune it (An et al., 2025; Liu et al., 2025), without incorporating trajectory feedback. Our work frames temperature as a learnable meta-policy that adapts exploration online, guided directly by trajectory advantages and likelihood to align LLM policy optimization.

**Meta-Policy in Reinforcement Learning.** Meta-policies have been studied in conventional RL. In hierarchical RL, the meta-policy acts as a high-level controller over low-level skills or options (Vezhnevets et al., 2017; Bacon et al., 2017; Frans et al., 2017). MLSH (Frans et al., 2017) learns a set of reusable sub-policies, which are shared across tasks, while a task-specific master policy then composes these sub-policies to solve new tasks. Another line of work leverages meta-gradient methods to treat hyperparameters—such as discount factors, bootstrapping parameter $\lambda$—as differentiable variables updated through meta-gradients (Xu et al., 2018; Wang & Ni, 2020). Meta-SAC learns the optimal entropy regularization parameter in Soft Actor-Critic (Wang & Ni, 2020).

While these works highlight the potential of meta-policies in conventional RL, they have not been explored in the context of LLM RL. To the best of our knowledge, our approach is the first to treat sampling temperature as a meta-policy, directly addressing the exploration–exploitation trade-off during LLM RL. Note that the hyperparameters studied in (Xu et al., 2018; Wang & Ni, 2020) are differentiable, enabling direct optimization. In contrast, the sampling temperature in the typical LLM RL frameworks is non-differentiable, rendering these methods infeasible. We propose the practical TAMPO approach, which efficiently adapts the sampling temperature by reusing existing rollouts without requiring gradient-based optimization.

## 3 TEMPERATURE ADAPTIVE META POLICY OPTIMIZATION (TAMPO)

We aim to treat temperature as a learnable meta-policy that dynamically balances exploration and exploitation during LLM reinforcement learning. Given a discrete set of candidate temperatures

$\mathcal{T} = \{T_1, \ldots, T_K\}$, we define a meta-policy $\pi(T)$ that outputs a probability distribution over temperatures. During training, each rollout is sampled at a temperature determined based on $\pi(T)$. To achieve this, we introduce Temperature Adaptive Meta-Policy Optimization (TAMPO), a hierarchical framework that shifts probability mass toward temperatures that produce high-advantage trajectories while suppressing ineffective ones.

In this section, we first review background in §3.1, then formalize the problem in §3.2, and finally present TAMPO in §3.3.

## 3.1 BACKGROUND

We consider an LLM parameterized by $\theta$, trained with reinforcement learning. For a given prompt $q$, the model generates a trajectory $\tau_i = (o_{i,1}, \ldots, o_{i,n})$ of $n$ tokens, where each token $o_{i,t}$ is generated based on the state $s_{i,t} = (q, o_{i,<t})$. Reward is provided at the sequence level as $r_i$.

**Critic-Free RL with GRPO.** In LLM RL, critic-free algorithms have become widely adopted due to their scalability and simplicity. One representative method is Group Relative Policy Optimization (GRPO) (Shao et al., 2024; Guo et al., 2025), which provides a simple way to compute trajectory-level credit/advantage without an explicit critic. For a given prompt, it samples a group of $G$ rollouts $\{\tau_1, \ldots, \tau_G\}$ at a given temperature $T$. It then computes trajectory-level **advantages** $A_i$ by normalizing rewards $r_i$ within the group:

$$A_i = \frac{r_i - \text{mean}(\{r_1, \ldots, r_G\})}{\text{std}(\{r_1, \ldots, r_G\})}. \tag{1}$$

GRPO updates the policy $\pi_{\theta,T}$ by maximizing the expected advantage while including a KL regularization term with respect to a reference policy (see Appendix A for the full objective).

**Temperature in Rollout Generation.** During rollout generation, the sampling distribution is controlled by a temperature parameter $T > 0$, which scales the token logits $z(o_{i,t} \mid s_{i,t})$ as:

$$\pi_{\theta,T}(o_{i,t} \mid s_{i,t}) = \frac{\exp(z(o_{i,t} \mid s_{i,t})/T)}{\sum_{o_{i,t}'} \exp(z(o_{i,t}' \mid s_{i,t})/T)}. \tag{2}$$

The temperature controls the exploration–exploitation trade-off: too low results in over-exploitation, too high introduces excessive randomness, reducing useful exploration. Despite its importance, $T$ is typically fixed, limiting policy optimization.

## 3.2 PROBLEM FORMULATION

We formalize temperature adaptation as a bilevel meta-optimization problem. The inner loop optimizes the LLM policy $\pi_\theta$ under temperatures sampled from the meta-policy $\pi_\phi$, while the outer loop optimizes $\pi_\phi$ to maximize the performance of the LLM policy.

Let $\pi_\theta(\cdot \mid q; T)$ denote the model policy under temperature $T$ (which we also denote as $\pi_{\theta,T}(\cdot \mid q)$), where $q \sim \mathcal{D}$ is a prompt. The meta-policy $\pi_\phi$ specifies a distribution over candidate temperatures. The inner objective is

$$\theta^\star(\phi) = \arg\max_\theta \mathbb{E}_{q \sim \mathcal{D}} \, \mathbb{E}_{T \sim \pi_\phi} \, \mathbb{E}_{\tau \sim \pi_\theta(\cdot \mid q; T)} \big[ r(\tau, q) \big], \tag{3}$$

while the outer objective seeks

$$\phi^\star = \arg\max_\phi \, J_{\text{meta}} \big( \theta^\star(\phi) \big), \tag{4}$$

where $J_{\text{meta}}$ denotes the evaluation metric of interest (*e.g.*, expected reward). This bilevel formulation highlights the adaptive role of temperature, but is not directly tractable in practice. Next, we describe our TAMPO.

## 3.3 TAMPO: MECHANISM AND IMPLEMENTATION

**Tractability Challenge.** In many LLM RL pipelines, rollout generation and policy optimization are typically decoupled: rollouts are generated using lower-precision models for efficiency, which

prevents backpropagation of gradients through the sampling process. This makes direct gradient-based optimization of the temperature meta-policy infeasible. Moreover, a naive trial-and-error strategy—generating rollouts for each candidate temperature—is computationally prohibitive. We thus seek a solution that works with the existing decoupled pipeline and requires no extra rollouts.

**Key Observation.** Every trajectory inherently encode its "preferred" temperature, *i.e.*, the temperature under which it is most likely to be generated (see Figure 2). Intuitively, for a high-reward (positive advantage) trajectory, we should reinforce temperatures that increase its likelihood; for a low-reward trajectory (negative advantage), we should down-weight such temperatures. *This insight provides a tractable signal for adapting temperature.*

Building on this idea, TAMPO treats temperature as a *learnable meta-policy* updated directly from trajectory-level signals. The meta-policy shifts probability mass toward temperatures associated with advantageous rollouts and suppresses ineffective ones, enabling adaptive exploration aligned with policy improvement.

### 3.3.1 TEMPERATURE-DEPENDENT TRAJECTORY LIKELIHOOD

The likelihood (*i.e.*, probability) of a trajectory $\tau_i$ at temperature $T$ from policy $\theta$ is

$$P_{\theta,T}(\tau_i) = \prod_{t=1}^{|\tau_i|} \pi_{\theta,T}(o_{i,t}|s_{i,t}). \qquad (5)$$

To remove dependence on trajectory length, we use the **average log-likelihood** (likelihood for short hereafter):

$$\ell_T(\tau_i) = \frac{1}{|\tau_i|} \log P_{\theta,T}(\tau_i)$$

$$= \frac{1}{|\tau_i|} \sum_{t=1}^{|\tau_i|} \log \pi_{\theta,T}(o_{i,t} \mid s_{i,t}). \qquad (6)$$

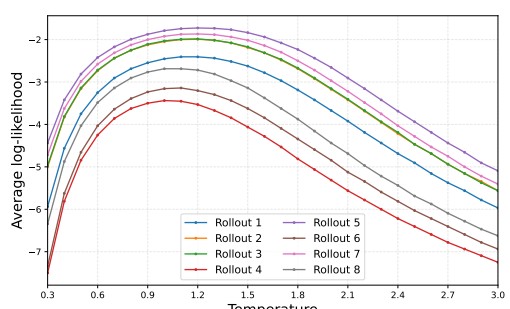

Figure 2: Example of trajectory likelihood under different temperatures for 8 rollouts of a prompt.

We can probe the likelihoods of a trajectory $\tau_i$ under a set of **virtual temperatures** $T_k \in \mathcal{T}$ via (6). The trajectory likelihood is **unimodal** w.r.t. $T$ (**see the proof in Appendix B**), and there exists a *likelihood-optimal temperature* $T_i^\star$ that maximizes the likelihood:

$$T^\star = \arg\max_{T_k \in \mathcal{T}} \ell_{T_k}(\tau_i). \qquad (7)$$

Figure 2 visualizes the trajectory likelihoods under different temperatures for 8 rollouts of a prompt, illustrating the unimodal nature. This implies that we can increase the likelihood of a trajectory by adjusting the temperature.

The empirical analysis in Appendix §C shows that positive advantage trajectories cluster to values $T^\star$ distinct from those of negative advantage trajectories, indicating that sampling from suitable temperatures can preferentially increase the likelihood of high advantage rollouts.

### 3.3.2 META-POLICY OPTIMIZATION WITH TEMPERATURE REWARDS

We formulate online temperature adaptation as a policy over temperature. A key challenge is evaluating the advantage of specific temperatures. Instead of sampling additional trajectories, we reuse the rollouts from inner-loop optimization. Particularly, for a trajectory with positive advantage, we increase its likelihood by rewarding temperatures that lead to higher trajectory likelihoods. Conversely, for a trajectory with negative advantage, we down-weight its likelihood by punishing temperatures that lead to higher trajectory likelihoods. This aligns temperature adaptation with policy optimization: reinforcing the likelihoods of high-reward trajectories, while suppressing likelihoods of low-reward trajectories.

**Temperature-specific Advantage.** Let $\tau_i$ denote a sampled trajectory and $\mathcal{T} = \{T_1, \ldots, T_K\}$ a set of candidate temperatures. For each trajectory $\tau_i$ and virtual candidate temperature $T_k$, similar to

(6), we compute the trajectory's likelihood under the LLM policy $\theta$ at temperature $T_k$ as

$$\ell_{T_k}(\tau_i) = \log P_{T_k}(\tau_i) = \sum_{t=1}^{|\tau_i|} \log \pi_{\theta, T_k}(o_{i,t} \mid s_{i,t}). \tag{8}$$

To capture the relative desirability of different temperatures, we normalize the likelihoods $\ell_{T_k}(\tau_i)$ across the $K$ candidate temperatures using *sparsemax* (Martins & Astudillo, 2016), yielding normalized likelihood $\hat{\ell}_{T_j}(\tau_i)$ with $\sum_{j=1}^{K} \hat{\ell}_{T_j}(\tau_i) = 1$. The trajectory's advantage $A_i$ (see (1)) is then scaled by these normalized likelihoods to produce the *temperature-specific advantage*:

$$\mathcal{A}_i^{(T_k)} = \hat{\ell}_{T_k}(\tau_i) \cdot A_i. \tag{9}$$

This can be interpretation as below:

- Positive-advantage trajectories ($A_i > 0$) reinforce the likelihood-optimal temperature most strongly, with neighboring temperatures receiving attenuated positive contributions.
- Negative-advantage trajectories ($A_i < 0$) penalize the likelihood-optimal temperature most, while nearby temperatures inherit attenuated negative contributions.

**Meta-policy Update.** Using trajectory's temperature-specific advantages, we maintain a meta-policy $\pi(T)$ over $\mathcal{T}$, which characterizes a probability distribution over $\mathcal{T}$. For a batch $\mathcal{B}$ of $|\mathcal{B}|$ samples, each with $G$ generated trajectories, we aggregate the temperature-specific advantages for each candidate temperature $T_k$:

$$\mathcal{A}_{\mathcal{B}}^{(T_k)} = \frac{1}{|\mathcal{B}|G} \sum_{b=1}^{|\mathcal{B}|} \sum_{i=1}^{G} \mathcal{A}_{b,i}^{(T_k)}, \tag{10}$$

where $\mathcal{A}_{b,i}^{(T_k)}$ denotes the temperature advantage of the $i$-th trajectory of sample $b$ with respect to temperature $T_k$. $\mathcal{A}_{\mathcal{B}}^{(T_k)}$ serves as the update target for the meta-policy, representing the batch-level temperature-specific advantage.

To stabilize updates, we maintain an exponentially weighted moving average (EMA) of temperature-specific advantages:

$$\bar{\mathcal{A}}_s^{(T_k)} = (1 - \alpha)\, \bar{\mathcal{A}}_{s-1}^{(T_k)} + \alpha\, \mathcal{A}_{\mathcal{B}}^{(T_k)}, \tag{11}$$

where $\alpha \in [0, 1)$ controls smoothing, $s$ denotes training step index. This EMA provides a stabilized advantage estimate, reducing variance from individual batches while retaining responsiveness to new trajectory feedback.

Finally, the meta-policy $\pi_s(T_k)$ at step $s$ is computed simply via min-max normalization:

$$\pi_s(T_k) = \frac{\tilde{\mathcal{A}}_s^{(T_k)}}{\sum_{j=1}^{K} \tilde{\mathcal{A}}_s^{(T_j)}}, \qquad \tilde{\mathcal{A}}_s^{(T_k)} = \frac{\bar{\mathcal{A}}_s^{(T_k)} - \min_j \bar{\mathcal{A}}_s^{(T_j)}}{\max_j \bar{\mathcal{A}}_s^{(T_j)} - \min_j \bar{\mathcal{A}}_s^{(T_j)}}, \quad k = 1, \ldots, K. \tag{12}$$

This ensures $\pi_s(T_k)$ forms a valid distribution: $\sum_{k=1}^{K} \pi_s(T_k) = 1$, favouring higher-advantage temperatures while suppressing lower-advantage ones, and forming a valid distribution: $\sum_{k=1}^{K} \pi_s(T_k) = 1$.

### 3.4 Overall Algorithm

Algorithm 1 summarizes the TAMPO, which operates as a **hierarchical two-loop process**:

- **Inner loop**: Optimizes the LLM policy using trajectories (*e.g.*, via GRPO) sampled under the temperature selected with meta-policy.
- **Outer loop**: Adaptively updates the temperature meta-policy. The same inner-loop trajectories are reused to update the meta-policy: trajectory advantages are modulated by their likelihoods under each candidate temperature as temperature-specific advantages (see Equations 8–9).

This **trajectory-guided update** enables principled online temperature adaptation, balancing exploration and exploitation **without extra rollouts**. TAMPO is compatible with critic-free RL algorithms and introduces negligible additional cost. We use GRPO as our critic-free RL in our experiments.

---

**Algorithm 1** Temperature Adaptive Meta Policy Optimization (TAMPO)

---

**Require:** LLM policy $\pi_\theta$, reference policy $\pi_{\text{ref}}$, training step number $S$, temperature candidates $\mathcal{T} = \{T_1, \ldots, T_K\}$, temperature meta-policy $\pi$, EMA decay $\alpha$.

1: Initialize $\bar{\mathcal{A}}_0^{(T_k)} \leftarrow 0$, $\pi_{s-1}(T_k) \leftarrow 1/K$ for all $T_k \in \mathcal{T}$.
2: **for** $s = 1$ to $S$ **do**
3:     **Outer Loop: Sample Temperature**
4:     Sample temperature $T_s \sim \pi_{s-1}(T)$.
5:
6:     **Inner Loop: Critic-free Policy Optimization**
7:     Sample a batch of prompts $\mathcal{B}$.
8:     **for** each prompt $q \in \mathcal{B}$ **do**
9:         Generate $G$ trajectories $\{\tau_1, \ldots, \tau_G\}$ using $\pi_\theta$ at temperature $T_s$.
10:        Compute reward $r(\tau_{q,i})$ and advantage $A_{q,i}$ (*e.g.*, via GRPO) for each trajectory $\tau_{q,i}$.
11:        Update LLM policy $\theta$ using advantages $A_{q,i}$.
12:     **end for**
13:
14:     **Outer Loop: Update Meta-Policy**
15:     Calculate temperature-specific advantages $\mathcal{A}_{\mathcal{B}}^{(T_k)}$ for all $T_k \in \mathcal{T}$ (see (10)).
16:     **for** each temperature $T_k \in \mathcal{T}$ **do**
17:        Update EMA: $\bar{\mathcal{A}}_s^{(T_k)} \leftarrow (1 - \alpha)\,\bar{\mathcal{A}}_{s-1}^{(T_k)} + \alpha\,\mathcal{A}_{\mathcal{B}}^{(T_k)}$.
18:     **end for**
19:     Update meta-policy $\pi_s(T_k)$ based on the $\bar{\mathcal{A}}_s^{(T_k)}$ and (12) for all $T_k \in \mathcal{T}$.
20: **end for**

---

## 4 EXPERIMENTS

### 4.1 SETUP OF MAIN EXPERIMENTS

**Training Datasets and Benchmarks.** We utilize a public math–reasoning dataset open-s1 (Dang & Ngo, 2025) for training. For evaluation, we use five distinct mathematics-focused benchmarks—**AIME24**, **MATH-500**, **AMC23**, **Minerva**, and **OlympiadBench**—covering a broad spectrum of difficulty levels and problem styles to comprehensively assess reasoning ability and generalization performance.

**Implementation Details.** We use DeepSeek-R1-Distill-Qwen-1.5B (DS-Qwen-1.5B for short) (Guo et al., 2025) as our base model to train both baselines and our models. We set our candidate temperatures in the range 0.6–1.5 with the interval 0.1, and $K = 10$, and the EMA coefficient $\alpha = 0.05$ by default. In the warmup phase (*i.e.*, the first 10% of training steps), we use a fixed sampling temperature of 1.0 for LLM policy, while the meta-policy is updated. After the warmup phase, the samping temperature is determined dynamically by the online learned meta-policy. The maximum response length is set to 6k tokens.

All experiments are conducted on NVIDIA $8 \times$V100 GPUs. We train both baseline models and our models for 200 steps, using an initial learning rate of $1 \times 10^{-6}$, a warmup ratio of $0.1$, followed by a cosine schedule. The training batch size is set to 32, with 8 rollouts per question.

**Evaluation Protocol.** We evaluate using Pass@1 and Pass@8 in order to measure single-shot accuracy and performance under multiple sampled attempts, showing model's exploration potential to solve questions. All evaluations are performed with a maximum response length of 6k tokens and a fixed decoding temperature of 1.0.

### 4.2 MAIN RESULTS

**Comparison with Baselines.** We adopt GRPO as the RL algorithm for both baselines and our scheme. DS-Qwen-1.5B refers to DeepSeek-R1-Distill-Qwen-1.5B, which serves as the starting model for all 1.5B model experiments. We construct baselines using a fixed sampling temperature at 0.9, 1.2, and 1.5 during training, denoted by GRPO ($T_s : 0.9$), GRPO ($T_s : 0.9$), and GRPO ($T_s :

| Method | Average | | AIME24 | | MATH-500 | | AMC23 | | Minerva | | OlympiadBench | |
|---|---|---|---|---|---|---|---|---|---|---|---|---|
| | Pass@1 | Pass@8 | Pass@1 | Pass@8 | Pass@1 | Pass@8 | Pass@1 | Pass@8 | Pass@1 | Pass@8 | Pass@1 | Pass@8 |
| DS-Qwen-1.5B | 39.1 | 57.8 | 13.3 | 26.7 | 76.2 | 89.2 | 45.0 | 72.5 | 22.8 | 41.5 | 38.4 | 59.0 |
| GRPO ($T_s$ : 0.9) | 42.0 | 60.8 | 20.0 | 30.0 | 75.2 | **91.0** | 50.0 | 80.0 | 26.1 | 43.4 | 38.7 | 59.4 |
| GRPO ($T_s$ : 1.2) | 41.6 | 61.1 | 20.0 | 33.3 | **77.4** | 90.6 | 50.0 | 77.5 | 22.4 | 43.4 | 38.1 | 60.5 |
| GRPO ($T_s$ : 1.5) | 42.6 | 62.1 | **23.3** | 36.7 | 75.4 | 90.8 | 52.5 | 77.5 | 22.8 | 44.5 | 39.0 | **61.2** |
| GRPO ($T_s$ : $0.9 \rightarrow 1.5$) | 42.8 | 59.7 | 16.7 | 30.0 | 76.6 | 89.8 | **55.0** | 77.5 | 24.6 | 40.8 | **41.0** | 60.4 |
| TAMPO (Ours) | **44.5** | **63.8** | **23.3** | **40.0** | 76.8 | 91.0 | **55.0** | **82.5** | **27.9** | **44.8** | 39.6 | 60.7 |

Table 1: Comparison of TAMPO with baselines on math reasoning using 1.5B models, evaluated with Pass@1 and Pass@8. DS-Qwen-1.5B denotes DeepSeek-R1-Distill-Qwen-1.5B (Guo et al., 2025), which serves as the base model for all training on the open-s1 dataset. GRPO ($T_s$ : 0.9) indicates a baseline trained with GRPO at a fixed sampling temperature of 0.9. The maximum response length is set to 6k tokens. **Best** results are in bold, and second-best results are underlined.

0.9), respectively. Additionally, we include a baseline with a linearly increasing training temperature from 0.9 to 1.5, denoted as GRPO ($T_s$ : $0.9 \rightarrow 1.5$).

Table 1 reports Pass@1 and Pass@8 accuracy across the five evaluation benchmarks. On average, our TAMPO outperforms the best fixed-temperature baseline, achieving 1.9% and 1.7% improvements in Pass@1 and Pass@8, respectively. Across all datasets, TAMPO either achieves the best performance or the second best, highlighting the effectiveness of treating temperature as a learnable meta-policy.

**Complexity.** Our scheme has nearly the same computational complexity compared to the baseline schemes. (i) The meta-policy model is extremely lightweight, as it only maintains a list of temperature advantages, introducing negligible overhead during training and being discarded at inference. (ii) TAMPO reuses rollouts from the inner loop, eliminating the need for generating additional trajectories. For the same number of training steps, both the baseline schemes using GRPO and our TAMPO use approximately the same training time (9 hours 54 minutes on an $8 \times$V100 GPU machine for 200 steps).

### 4.3 ABLATION STUDY

**Influence of EMA Coefficient $\alpha$.** As described in (11), $\alpha$ controls the exponential moving average smoothing, balancing the contribution of current feedback and historical accumulation. Table 2 reports the results for $\alpha \in \{0.01, 0.05, 0.10\}$. The increase in $\alpha$ from 0.01 to 0.05 increases the average score from 41.6 to 44.5 and all benchmarks showing improvement. Further increasing $\alpha$ to 0.10 produces mixed effects. $\alpha = 0.05$ provides a good trade-off between reducing variance and retaining responsiveness to new feedback.

| $\alpha$ | Average | AIME24 | MATH-500 | AMC23 | Minerva | OlympiadBench |
|---|---|---|---|---|---|---|
| 0.01 | 41.6 | 20.0 | 75.2 | 50.0 | 25.4 | 37.5 |
| 0.05 | **44.5** | **23.3** | **76.8** | 55.0 | **27.9** | **39.6** |
| 0.10 | 43.6 | **23.3** | 75.4 | **57.5** | 23.2 | 38.8 |

Table 2: Influence of the EMA coefficient $\alpha$ for our TAMPO. We report the performance on Pass@1.

**Influence of Sampling Strategy on Meta-policy.** We model the meta-policy as a distribution $\pi$ over candidate temperatures. At each training step, similar to token sampling in LLM, we sample temperatures from $\pi$ via nucleus sampling (*i.e.*, top-p sampling), which selects actions from the smallest set whose cumulative probability exceeds a threshold $p$, focusing on the "nucleus" of likely outcomes while filtering out low-probability options. $p$ controls the trade-off between exploration and exploitation in the meta-policy.

Table 3 shows results for top-p sampling with $p = 0.9, 0.7, 0.5, 0$. When $p = 0$, this corresponds to greedy sampling, selecting the temperature with the highest probability (equivalent to top-k sampling with $k = 1$). We can see that a moderate threshold $p = 0.7$ achieves a good balance between exploration and exploration on temperature. We set $p$ to 0.7 by default. Too high a threshold lim-

| Sampling strategy | Average | AIME24 | MATH-500 | AMC23 | Minerva | OlympiadBench |
|---|---|---|---|---|---|---|
| Top-p ($p$: 0.9) | 43.0 | 20.0 | 76.6 | 52.5 | 26.1 | **39.7** |
| Top-p ($p$: 0.7) | **44.5** | **23.3** | **76.8** | **55.0** | **27.9** | 39.6 |
| Top-p ($p$: 0.5) | 42.2 | **23.3** | 75.4 | 50.0 | 24.3 | 38.1 |
| Top-p ($p$: 0, greedy) | 40.9 | 20.0 | 73.8 | 50.0 | 23.5 | 37.2 |

Table 3: Influence of sampling strategy on meta-policy. We report the performance on Pass@1.

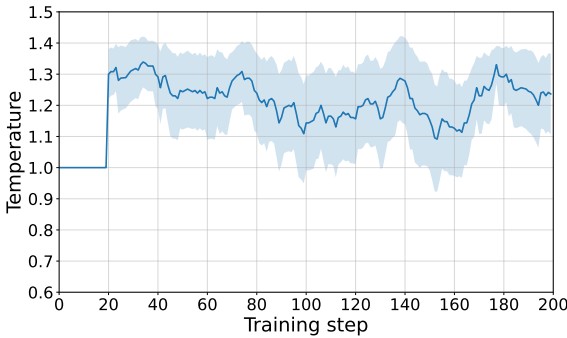

Figure 3: Sampling temperatures from the meta-policy using nucleus sampling.

its exploitation of the learned meta-policy, sacrificing certain opportunities to exploit the benefit of learned meta-policy. Too low a threshold (*e.g.*, $p = 0.5$) or greedy sampling (*i.e.*, $p = 0$) leads to under-exploration, reducing effectiveness. Since the greedy sampling always sample temperatures that having the highest advantages, even though the second-best temperature may have similar advantages, such under-exploration of temperatures leads to LLM policy losing opportunity to generate rollouts under those temperatures and reducing the experience diversity.

## 4.4 ANALYSIS

**Sampling Temperature from Meta-policy.** Figure 3 shows the temperatures sampled by the meta-policy using nucleus sampling ($p$=0.7), reporting the mean and standard deviation over a sliding window of 25 training steps. After 20 warm-up steps, the meta-policy favors a high temperature (around 1.3), encouraging exploration in LLM policy generation. As training proceeds, the mean temperature gradually decreases, shifting toward exploitation while maintaining diversity. The large variance arises from nucleus sampling, which balances exploitation of the learned meta-policy with exploration over diverse temperatures.

## 4.5 MORE EXPERIMENTS RESULTS

To evaluate TAMPO's effectiveness across different base models and domains, we conducted additional experiments. Specifically, we implemented TAMPO on Qwen2.5-3B-Instruct (Yang et al., 2024) base model and evaluated on ECQA (Aggarwal et al.), a commonsense reasoning benchmark. The results in Table 4 show consistent improvements, demonstrating that TAMPO generalizes beyond mathematical reasoning.

| Method | Pass@1 | Pass@8 |
|---|---|---|
| Qwen2.5-3B-Instruct (no RL) | 73.06% | 77.76% |
| GRPO | 75.07% | 78.94% |
| **TAMPO** | **76.12%** | **79.67%** |

Table 4: ECQA results evaluated with Pass@1 and Pass@8.

## 5 CONCLUSION

We presented TAMPO, a hierarchical framework that treats temperature as a learnable meta-policy in LLM reinforcement learning. By reusing trajectories to adaptively update a temperature meta-policy, TAMPO enables exploration without extra rollouts and achieves consistent improvements on mathematical reasoning benchmarks. These results demonstrate that principled temperature adaptation is a practical and effective tool for advancing LLM reinforcement learning.

## LLM USAGE

We used large language models (LLMs) to assist with refining the writing and presentation of this paper. LLMs were employed for improving clarity, conciseness, and formatting, while all ideas, methods, and experiments were conceived and executed by the authors.

## REPRODUCIBILITY STATEMENT

We have made extensive efforts to ensure the reproducibility of our work. Our temperature adaptation method and algorithm are described in § 3.3. The training details, including base models, hyperparameters, and optimization settings, are described in § 4.1. The training datasets and evaluation benchmarks used in this study are publicly available. We will release the source code upon acceptance to further facilitate the verification and reproduction of our results.

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

## A  GRPO Optimization Objective

GRPO updates the policy $\pi_{\theta,T}$ by maximizing the expected advantage while including a KL regularization term with respect to a reference policy:

$$\mathcal{J}_{\text{GRPO}}(\theta) = \mathbb{E}_{(q,a)\sim\mathcal{D},\{\tau_i\}_{i=1}^G \sim \pi_{\theta_{\text{old}},T}(\cdot|q)}$$

$$\left[ \frac{1}{G}\sum_{i=1}^{G}\frac{1}{|\tau_i|}\sum_{t=1}^{|\tau_i|}\left(\left(\rho_{i,t}(\theta,T)A_i, \text{clip}(\rho_{i,t}(\theta,T),1-\epsilon,1+\epsilon)A_i\right) - \beta D_{\text{KL}}(\pi_{\theta,T}||\pi_{\text{ref}})\right)\right], \tag{13}$$

where $\rho_{i,t}(\theta,T) = \frac{\pi_{\theta,T}(o_{i,t}|q,o_{i,<t})}{\pi_{\theta_{\text{old}},T}(o_{i,t}|q,o_{i,<t})}$. $\epsilon$ and $\beta$ are hyperparameters.

## B  Proof of the Unimodal of Trajectory Likelihood w.r.t. $T$

For a trajectory $\tau = (a_1,\ldots,a_n)$ of $n$ tokens, each token $a_t$ is generated based on the state $s_t = (q, a_{<t})$ with policy model parameterized by $\theta$. Here $q$ denotes the prompt/question. Under temperature scaling of $T > 0$, the conditional distribution can be obtained based on the model output logits $z(a \mid s_t)$ as

$$\pi_{\theta,T}(a \mid s_t) = \frac{\exp\left(z(a \mid s_t)/T\right)}{\sum_{a'}\exp\left(z(a' \mid s_t)/T\right)}. \tag{14}$$

The log-likelihood of the trajectory $\tau = (a_1, a_2, \ldots, a_n)$ is

$$\ell_T(\tau) = \log P_{\theta,T}(\tau), \qquad \text{where } P_{\theta,T}(\tau) = \prod_{t=1}^{n}\pi_{\theta,T}(a_t \mid s_t). \tag{15}$$

We define a baseline distribution at $T=1$ under condition $s_t$ as [1]

$$q_t(a) := \frac{e^{z(a|s_t)}}{\sum_u e^{z(u|s_t)}}. \tag{16}$$

Then,

$$\log q_t(a) = z(a \mid s_t) - \log\sum_u e^{z(u|s_t)}. \tag{17}$$

Let $\beta = 1/T$ which denotes the inverse temperature. We have

$$\pi_{\theta,T}(a \mid s_t) = \frac{q_t(a)^\beta}{\sum_u q_t(u)^\beta}. \tag{18}$$

Let us define $S(\tau) := \sum_{t=1}^{n}\log q_t(a_t)$, $A_t(\beta) := \log\sum_u q_t(u)^\beta$, and $A(\beta) := \sum_{t=1}^{n}A_t(\beta)$. We obtain the standard exponential family form:

$$\log P_{\theta,\beta}(\tau) = \beta\,S(\tau) - A(\beta). \tag{19}$$

**Second derivative w.r.t the inverse temperature $\beta$ (unimodality):** Based on standard properties,

$$A'_t(\beta) = \mathbb{E}_{a\sim\pi_{\theta,T}(\cdot|s_t)}[\log q_t(a)], \qquad A''_t(\beta) = \text{Var}_{a\sim\pi_{\theta,T}(\cdot|s_t)}[\log q_t(a)] \geq 0. \tag{20}$$

Then,

$$\frac{d^2}{d\beta^2}\log P_{\theta,\beta}(\tau) = -A''(\beta) \leq 0. \tag{21}$$

Therefore, $\log P_{\theta,\beta}(\tau)$ is strictly concave in $\beta$. Since $\beta = 1/T$ is a monotone reparameterization, $P_{\theta,T}(\tau)$ is unimodal in $T$, with a one-to-one correspondence between maximizers: $\beta^*(\tau) = 1/T^*(\tau)$.

---

[1] We denote $q_t(a_t|s_t)$ by $q_t(a_t)$ for conciseness.

**First derivative w.r.t. temperature and endpoint limits (typical "rise-then-fall"):** We have

$$\log \pi_{\theta,T}(a \mid s_t) = \frac{z(a \mid s_t)}{T} - \log \sum_u \exp(z(u \mid s_t)/T). \tag{22}$$

Let us define $\mu_T(s_t) := \mathbb{E}_{a \sim \pi_{\theta,T}(\cdot \mid s_t)}[z(a \mid s_t)]$. A direct differentiation yields

$$\frac{d}{dT} \log \pi_{\theta,T}(a \mid s_t) = -\frac{1}{T^2}\Big[z(a \mid s_t) - \mu_T(s_t)\Big]. \tag{23}$$

Therefore,

$$\ell'_T(T) = \frac{d}{dT} \log P_{\theta,T}(\tau) = -\frac{1}{T^2} \sum_{t=1}^{n}\Big[z(a \mid s_t) - \mu_T(s_t)\Big]. \tag{24}$$

As $T \to 0^+$, each step tends to select the action with maximum logit. Therefore, $\mu_T(s_t) \to \max_a z(a \mid s_t)$. When the trajecotry is not composed by greedy sampling of actions/tokens, $z(a|s_t) < \mu_T(s_t)$ and then $\ell'_T(T) > 0$, *i.e.*, a small increase of $T$ increases $\log P_{\theta,T}(\tau)$.

As $T \to \infty$, $\pi_{\theta,T}$ approaches uniform distribution and $\mu_T(s_t) \to \overline{z}(s_t) = |\mathcal{V}|^{-1} \sum_a z(a \mid s_t)$. The LLM sampled tokens in general have $z(a|s_t) > \overline{z}(s_t)$ (except for an edge case), making the sum positive and $\ell'_T(T) < 0$, *i.e.*, a increase of $T$ reduces $\log P_{\theta,T}(\tau)$.

**Conclusion: unimodality and "rise-then-fall".** Combining the strict concavity w.r.t. $\beta$ (Eq. (21)) and the opposite signs of $\ell'_T(T)$ at the low- and high-temperature limits (Eq. 24), we conclude that, for trajectories except edge cases, there exists a unique $T^*(\tau) \in (0, \infty)$ such that

$$\ell'_T(T) > 0 \; for \; T < T^*, \qquad \ell'_T(T^*) = 0, \quad and \; \ell'_T(T) < 0 \; for \; T > T^*. \tag{25}$$

Log-likelihood $\log P_{\theta,T}(\tau)$ is *typically* unimodal w.r.t $T$ and exhibits a "rise-then-fall" shape beside edge cases.

**Edge cases.** First, if a trajectory is composed of greedy tokens, that is, at every step $t$ LLM policy picks the most probable token, then its likelihood decreases monotonically with $T$, approaching 1 as $T \to 0^+$. Second, if a trajectory has many chosen tokens having probability below $|\mathcal{V}|^{-1}$, *i.e.*, which rarely occur under the decoding strategy, its likelihood will slowly increase when $T$ increases.

The unimodal property of the trajectory log-likelihood for general cases, and the monotonicity for edge cases, all assure that we can use discrete temperatures (as virtual temperatures) to well estimate the likelihood of trajectories under different temperatures.

## C  CONNECTION BETWEEN TEMPERATURE AND ADVANTAGE

To gain a more intuitive understanding of the correlation between trajectory advantage and likelihood-optimal temperature, we conducted training under fixed sampling temperatures (0.9, 1.2, and 1.5) using GRPO and computed the likelihood-optimal temperature $T^*(\tau)$ for each trajectory $\tau$. Figure 4 shows the resulting distributions for trajectories with positive advantages (green) and negative advantages (red) along training, with mean and variance computed for rollouts of every 5 training steps.

Three patterns can be observed. (i) A fixed sampling temperature generally does not align with the likelihood-optimal temperatures observed during training. (ii) Positive-advantage trajectories cluster around distinct $T^*$ values compared to negative-advantage trajectories, indicating that high-reward rollouts are intrinsically associated with specific temperature regimes. (iii) At extreme sampling temperatures, trajectories tend to revert toward moderate $T^*$ values: when $T = 1.5$, most $T^*$ are well below 1.5, while at $T = 0.9$, most $T^*$ are well above 0.9. This demonstrates that both excessive randomness and excessive determinism reduce the likelihood of producing advantageous rollouts.

Overall, these results suggest that there exist more optimal temperatures that increase the probability of high-advantage trajectories, highlighting substantial room for improving the likelihood of sampling advantageous rollouts.

## D  SYSTEM PROMPT

Figure 5 shows the system prompt used during our training.

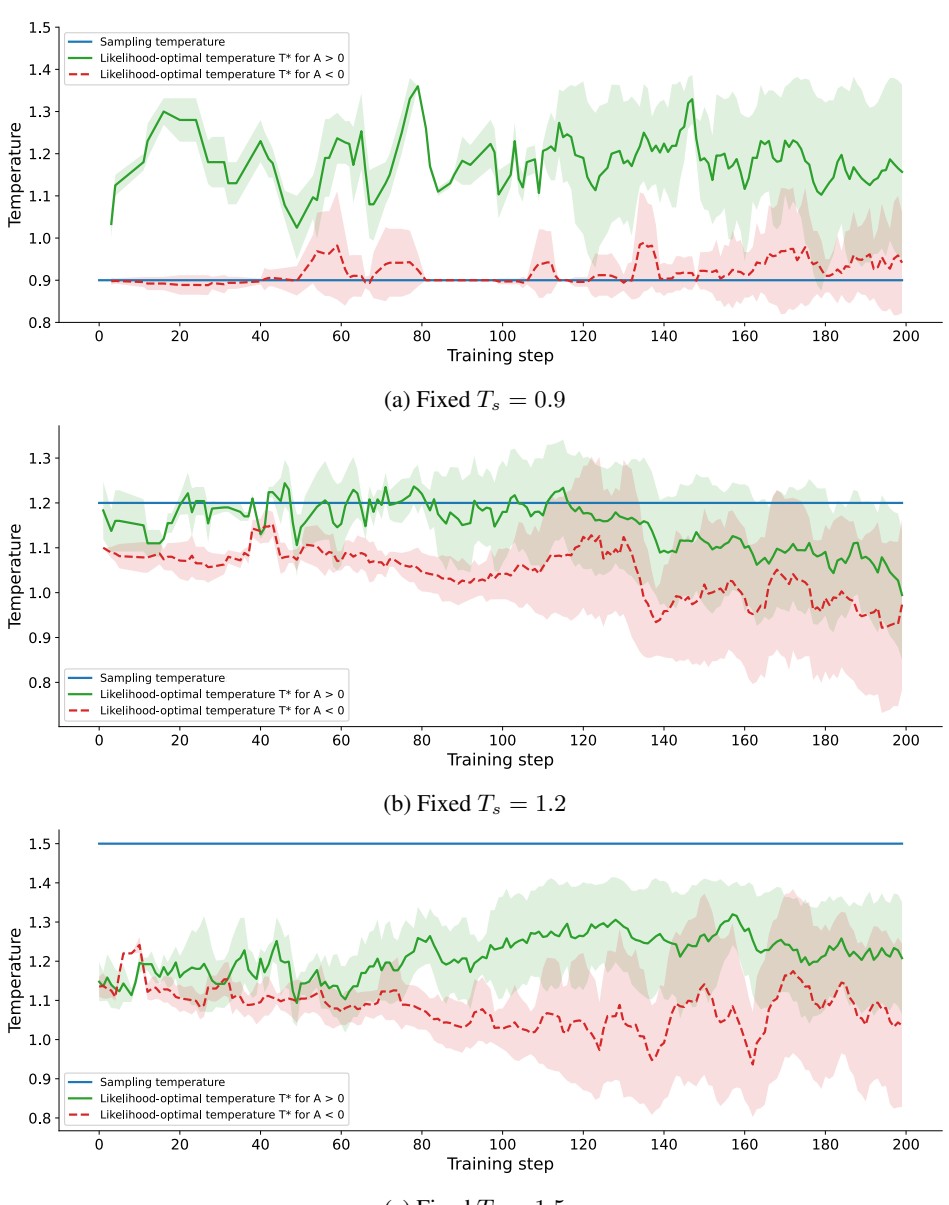

(a) Fixed $T_s = 0.9$

(b) Fixed $T_s = 1.2$

(c) Fixed $T_s = 1.5$

Figure 4: Distribution of trajectory likelihood-optimal temperatures under three fixed training temperatures, respectively. Green curve corresponds to the likelihood-optimal temperatures of positive-advantage trajectories ($A > 0$), red curve to negative-advantage trajectories ($A < 0$), and blue curve to the sampled fixed temperature.

```
A conversation between User and Assistant. The user asks a question,
and the Assistant solves it. The assistant first thinks about the
reasoning process in the mind and then provides the user with the
answer. The reasoning process and answer are enclosed within <think>
</think> and <answer> </answer> tags, respectively, i.e., <think>
[REASONING PROCESS HERE] </think> Therefore, the final answer is:
<answer> $\boxed{{ANSWER CONTENT HERE}}$ </answer>
```

Figure 5: System prompt for the policy model.

