# OpenReview forum: "Temperature as a Meta-Policy: Adaptive Temperature in LLM Reinforcement Learning"
_ICLR.cc/2026/Conference — ICLR 2026 Poster_

### Official Review · Reviewer_aLbY · 2025-10-28

**Soundness:** 2
**Presentation:** 2
**Contribution:** 2
**Rating:** 6
**Confidence:** 4

**Summary:**

This paper proposes an intuitive method TAMPO, which replacing the conventional static or heuristic temperature to a meta-policy and learning it online, thereby achieving trajectory-driven exploration–exploitation adaptation in LLM reinforcement learning. In detail, the framework consists two loop process: inner loop generates trajectories under temperatures determined by meta-policy and optimized (using GRPO) LLM, and outer loop optimize meta-policy based on temperature-specific advantage, which can be calculated by reused trajectories likelihood from inner loop under certain temperature. Experiments on mathematical reasoning tasks show robust improvements over several fixed or linearly scheduled baselines, along with ablations and analyses (e.g., temperature trajectories, T* distributions), demonstrating the potential and interpretability of the proposed method.

**Strengths:**

S1: Treating temperature as a meta-policy is a fresh and practical perspective in the context of LLM RL. It addresses a real limitation of current methods that use fixed or heuristically scheduled temperatures.
S2: TAMPO can be seamlessly combined with existing critic-free RL methods (such as GRPO and REINFORCE++) without adding additional sampling costs, and has good practicality and promotion potential.
S3: The paper is well-written and easy-to-follow.

**Weaknesses:**

W1: The current meta-policy employs a shared temperature distribution across the entire dataset, without explicit conditioning on the prompt, subtask, or training stage. This may lead to inefficiency in multi-task settings or under highly heterogeneous prompts.
W2: As the reinforce learning is known by its high variance, report the mean +- standard deviation along with significance tests under at least 3-5 independent runs is preferred to demonstrate the effectiveness of the methods.
W3: All experiments are conducted on a 1.5B parameter model and focus solely on mathematical reasoning tasks. It is unclear how well the method generalizes to larger models or other types of tasks (e.g., code generation, dialogue, summarization).
W4: Typo: Line 356 and 357, “using initial learning rate 1x10e-6” but “minimum learning rate of 0.1” is confusing and counterfactual.
W5: The range (0.6-1.5) and interval (0.1) of the candidate temperature set are manually preset without an adaptive adjustment mechanism. Moreover, the applicability of this fixed candidate set to different models or task scenarios is not verified, which may lead to missing better exploration intervals due to the limitations of the temperature search space.
W6: The trajectory likelihood relied on for meta-policy updates is strongly coupled with the dynamically iterated LLM policy in the inner loop. No mitigation measures are designed for the likelihood estimation bias caused by policy updates, which may result in the evaluation of temperature-specific advantages being affected by the current policy bias, reducing the stability of temperature adaptation.

**Questions:**

Please refer to Weakness.

**Details Of Ethics Concerns:**

N.A.

---

> ### Author Response · Authors · 2025-11-21
>
> We thank the reviewer for the thoughtful feedback and detailed suggestions. We also appreciate the recognition of TAMPO’s novelty in treating temperature as a meta-policy, its efficient integration with critic-free RL methods without additional rollout costs, and the clarity of presentation. Below we address the main points raised.
>
> ---
> Q1: The current meta-policy employs a shared temperature distribution across the entire dataset, without explicit conditioning on the prompt, subtask, or training stage. This may lead to inefficiency in multi-task settings or under highly heterogeneous prompts.
>
> A1: Thank you for the insightful comment. In the current TAMPO implementation, the meta-policy implicitly adapts to the training stage as the inner-loop policy evolves over time. Extending the meta-policy to explicitly condition on prompts, subtasks, or other task-specific features is a promising direction for multi-task or highly heterogeneous settings, and we leave this for future work.
>
> ---
> Q2: As the reinforce learning is known by its high variance, report the mean and standard deviation along with significance tests under at least 3-5 independent runs is preferred to demonstrate the effectiveness of the methods.
>
> A2: We conducted three independent runs and reported the mean and standard deviation to more reliably demonstrate the effectiveness of our method.
> Particularly, we trained TAMPO using three independent seeds and computed the mean and standard deviation across all five benchmarks (AIME24, MATH-500, AMC23, Minerva, OlympiadBench) for both Pass@1 and Pass@8.
>
> As shown in the table below, the overall deviations are small (e.g., overall Pass@1: 44.5 ± 0.15), and the results closely match those from a single run.
> We have added the results in our revised Appendix E.
>
> Mean ± Std over 3 runs
> | Metric Summary | Avg Pass@1 | Avg Pass@8 | AIME24 Pass@1 | AIME24 Pass@8 | MATH-500 Pass@1 | MATH-500 Pass@8 | AMC23 Pass@1 | AMC23 Pass@8 | Minerva Pass@1 | Minerva Pass@8 | OlympiadBench Pass@1 | OlympiadBench Pass@8 |
> |----------------|------------|------------|----------------|----------------|------------------|------------------|---------------|---------------|------------------|------------------|-------------------------|-------------------------|
> | **Mean ± Std (3 runs)** | \(44.5 ± 0.15\) | \(63.7 ± 0.12\) | \(22.2 ± 1.91\) | \(38.9 ± 1.91\) | \(77.0 ± 0.53\) | \(90.9 ± 0.12\) | \(55.8 ± 1.44\) | \(83.3 ± 1.44\) | \(27.8 ± 0.95\) | \(45.0 ± 0.57\) | \(39.6 ± 0.36\) | \(60.3 ± 0.40\) |
>
>
> ---
> Q3: Generalization to larger models and non-math domains.
>
> A3: We conducted additional experiments to evaluate TAMPO’s generality across different base models and domains. Specifically, we used Qwen2.5-3B-Instruct as a new base model and evaluated TAMPO on ECQA, a commonsense reasoning benchmark.
>
> The results, summarized below, show consistent improvements, demonstrating that TAMPO generalizes beyond mathematical reasoning. These results have been added to the revised manuscript in Appendix E.
>
> ECQA Results
> | Method                         | Pass@1  | Pass@8  |
> |--------------------------------|---------|---------|
> | Qwen2.5-3B-Instruct (no RL)    | 73.06% | 77.76% |
> | GRPO                           | 75.07% | 78.94% |
> | **TAMPO**                      | **76.12%** | **79.67%** |

---

> ### Author Response · Authors · 2025-11-21
>
> Q4: Typo: Line 356 and 357, “using initial learning rate 1x10e-6” but “minimum learning rate of 0.1” is confusing and counterfactual.
>
> A4: Thank you very much for pointing out this typo. We corrected it to:
> "using an initial learning rate of $1 \times 10^{-6}$, a warmup ratio of 0.1, followed by a cosine schedule." in the revision.
>
> ---
> Q5: The range (0.6-1.5) and interval (0.1) of the candidate temperature set are manually preset without an adaptive adjustment mechanism. Moreover, the applicability of this fixed candidate set to different models or task scenarios is not verified, which may lead to missing better exploration intervals due to the limitations of the temperature search space.
>
> A5: Thank you for the comment. In future work, we plan to explore mechanisms for automatically expanding or shrinking the temperature range to reduce manual specification and better adapt to different models and task scenarios. Currently, the range [0.6, 1.5] is chosen based on common practices in LLM RL, and the same range was used in the experiments described in A3.
>
> Empirically, expanding the grid to a finer interval (e.g., 50 candidate temperatures) yields no additional performance improvement, as shown below. This indicates that the current granularity is sufficient. Note that the trajectory likelihood varies smoothly and is unimodal with respect to temperature, so this discretization provides a reliable approximation.
>
> | Method | Avg Pass@1 | Avg Pass@8 | AIME24 Pass@1 | AIME24 Pass@8 | MATH-500 Pass@1 | MATH-500 Pass@8 | AMC23 Pass@1 | AMC23 Pass@8 | Minerva Pass@1 | Minerva Pass@8 | OlympiadBench Pass@1 | OlympiadBench Pass@8 |
> |--------|------------|------------|----------------|----------------|------------------|------------------|---------------|---------------|------------------|------------------|-------------------------|-------------------------|
> | TAMPO with 50 temperatures | 44.6 | 63.6 | 23.3 | 36.7 | 76.6 | 90.8 | 55.0 | 82.5 | 28.7 | 45.6 | 39.1 | 60.2 |
>
>
> ---
> Q6: The trajectory likelihood relied on for meta-policy updates is strongly coupled with the dynamically iterated LLM policy in the inner loop. No mitigation measures are designed for the likelihood estimation bias caused by policy updates, which may result in the evaluation of temperature-specific advantages being affected by the current policy bias, reducing the stability of temperature adaptation.
>
> A6: Thank you for this insightful comment. We acknowledge that temperature-specific advantages are estimated using trajectory likelihoods produced by the evolving inner-loop policy, which can introduce bias when the rollout distribution drifts.
>
> In practice, we find TAMPO robust for two reasons:
> 1.	Batch and temporal aggregation. The outer-loop update aggregates likelihoods across batches and over training steps, reducing variance and smoothing away short-term policy-induced bias.
> 2.	Smooth temperature–likelihood relationship. Trajectory likelihood as a function of temperature is stable and smooth (as shown in Fig. 2), making the meta-policy update less sensitive to local estimation noise.

---

### Official Review · Reviewer_a4qg · 2025-10-30

**Soundness:** 2
**Presentation:** 3
**Contribution:** 2
**Rating:** 4
**Confidence:** 3

**Summary:**

This paper proposes Temperature Adaptive Meta Policy Optimization (TAMPO), a new framework that recasts temperature control in large language model (LLM) reinforcement learning (RL) as a learnable meta-policy rather than a fixed hyperparameter.
In typical LLM RL (e.g., GRPO or PPO-based RLHF), temperature controls the exploration–exploitation balance during text generation.
However, most existing methods use static or heuristic temperature schedules, which fail to adapt to the evolving training dynamics.
TAMPO formulates temperature control as a two-level learning process:
In the inner loop, the LLM policy is optimized using critic-free RL (e.g., GRPO) at the temperature chosen by the meta-policy.
In the outer loop, the meta-policy updates its temperature based on temperature-specific advantages by reusing the inner loop rollouts, reinforcing those that are more likely to generate trajectories with high advantage trajectories, without requiring additional rollouts.

Experiments on five distinct mathematics-focused benchmarks (AIME24, MATH-500, AMC23, Minerva, and OlympiadBench) show that TAMPO outperforms fixed or heuristic temperature baselines, in terms of Pass@1 and Pass@8 accuracy, by approximately 1.9% and 1.7%, respectively.

**Strengths:**

- This work provides a interesting way to control exploration in LLM RL by recasting temperature as a meta-policy variable.
- TAMPO efficiently reuses existing rollouts for meta-policy updates, introducing negligible additional cost.
- TAMPO demonstrates consistent improvements across multiple reasoning benchmarks using DeepSeek-R1-Distill-Qwen-1.5B.

**Weaknesses:**

- Experiments are restricted to mathematical reasoning tasks $\to$ generalization to other domains (dialogue, code generation, ...) remains untested.
- It’s unclear how TAMPO can be applied to critic-based or hybrid RLHF methods.
- The approach uses a fixed discrete set of temperatures {0.6, 0.7, ..., 1.4, 1.5}. Continuous temperature optimization might yield smoother adaptation.
- The paper does not include an ablation study for different base models.
- While results are strong, interpretability and deeper insight into how adaptive temperature affects reasoning quality are somewhat limited.

**Questions:**

- It would be helpful to know whether the authors have conducted any ablation studies to assess the generality of TAMPO:
  1. Using base models other than DeepSeek-R1-Distill-Qwen-1.5B,
  2. Across different domains such as dialogue or code generation, and
  3. With alternative critic-free reinforcement learning algorithms.
- Could the authors clarify what the optimal solution of the iterative optimization represents, both from a theoretical and an intuitive perspective?
- Could the authors elaborate on whether they plan to extend TAMPO to continuous temperature optimization rather than relying on a fixed discrete set of candidates?
- Why does adaptive temperature lead to improved reasoning quality? In addition, are there any potential negative effects of adaptive temperature, such as reduced exploration or excessive exploration?

---

> ### Author Response · Authors · 2025-11-21
>
> We thank the reviewer for the thoughtful feedback and detailed suggestions. We also appreciate the reviewer’s recognition of TAMPO’s novelty in recasting temperature as a meta-policy, its efficient reuse of rollouts for outer-loop updates with negligible additional cost, and the consistent improvements it achieves across multiple reasoning benchmarks. Below, we address the main points raised.
>
>
> ---
> Q1: Generalization to other base models/domains
>
> A1: We conducted additional experiments to evaluate TAMPO’s generality across different base models and domains. Specifically, we used Qwen2.5-3B-Instruct as a new base model and evaluated TAMPO on ECQA, a commonsense reasoning benchmark.
>
> The results, summarized below, show that TAMPO consistently improves performance, demonstrating that its effectiveness does not rely on a specific policy model and extends beyond mathematical reasoning. These results have been added to the revised manuscript.
>
>
> | Method                         | Pass@1  | Pass@8  |
> |--------------------------------|---------|---------|
> | Qwen2.5-3B-Instruct (no RL)    | 73.06% | 77.76% |
> | GRPO                           | 75.07% | 78.94% |
> | **TAMPO**                      | **76.12%** | **79.67%** |
>
>
> ---
> Q2: Could the authors clarify what the optimal solution of the iterative optimization represents, both from a theoretical and an intuitive perspective?
>
> A2: The meta-policy online learns temperature distributions that maximize expected advantages of rollouts. Intuitively, this corresponds to selecting temperatures that most reliably generate high-quality trajectories, achieving an optimal trade-off between correctness and exploration at the current training stage. In other words, the iterative optimization learns which temperature is most effective for producing advantageous rollouts.
>
> Theoretically, the outer-loop optimization seeks the temperature $T^*$ that maximizes the expected temperature-specific advantage over trajectories:
>
> $$
> T^* = \arg\max_{T_k \in \mathcal{T}} \mathbb{E}\_{\tau_i \sim \pi_\theta} \big[ \mathcal{A}\_i^{(T_k)} \big]
> = \arg\max_{T_k \in \mathcal{T}} \mathbb{E}\_{\tau_i \sim \pi_\theta} \big[ \hat{\ell}_{T_k}(\tau_i) \cdot A\_i (\tau_i)\big],
> $$
>
> where $\hat{\ell}_{T_k}(\tau_i)$ is the normalized likelihood of trajectory $\tau_i$ under temperature $T_k$, and $A_i$ is the standard advantage. Positive-advantage trajectories reinforce temperatures that favor their likelihood, while negative-advantage trajectories suppress them, with neighboring temperatures receiving proportionally weighted contributions.
>
> ---
> Q3: Could the authors elaborate on whether they plan to extend TAMPO to continuous temperature optimization rather than relying on a fixed discrete set of candidates?
>
> A3: Thank you for the question. In the current TAMPO framework, we employ a discrete set of candidate temperatures to efficiently reuse inner-loop rollouts for meta-policy updates. This discrete design provides a good balance between efficiency and precision, as the trajectory likelihood is smooth and unimodal, and experiments show that finer discretization yields negligible gains.
>
> Extending TAMPO to continuous temperature optimization is an interesting future direction. One possible approach is to fit a continuous curve representing the trajectory likelihood across temperatures and perform gradient-based updates on the temperature parameter. This would require designing suitable fitting functions and adapting them for different rollouts. We view TAMPO as a practical and efficient solution for adaptive temperature learning, while continuous extensions could further improve flexibility in future work.

---

> ### Author Response · Authors · 2025-11-21
>
> Q4: Why does adaptive temperature lead to improved reasoning quality? In addition, are there any potential negative effects of adaptive temperature, such as reduced exploration or excessive exploration?
>
> A4: Intuitively, TAMPO aims to increase the likelihood of collecting high-advantage / high-quality trajectories during training. Since the sampling temperature governs the exploration–exploitation balance in LLMs, dynamically adjusting it allows the model to align exploration intensity with the current state of the policy more effectively than a fixed temperature.
>
> Mechanically, the outer-loop update in TAMPO maximizes the temperature-specific advantage (see A2). For positive-advantage trajectories, we reinforce their likelihood by encouraging temperatures that increase their probability. Conversely, for negative-advantage trajectories, we reduce their likelihood by discouraging temperatures that increase their probability.
>
> This mechanism allows TAMPO to adaptively guide exploration toward promising regions of the temperature space, improving reasoning quality while avoiding overly narrow or excessive exploration by leveraging feedback from trajectory advantages.
>
> ---
> Q5: Applicability to critic-based or hybrid RLHF methods
>
> A5: Thank you for the question. TAMPO is theoretically compatible with critic-based and hybrid RLHF frameworks. In such settings, the trajectory advantage can be computed using a critic: For each trajectory, the critic provides step-wise or trajectory-level advantage estimates, which can then be converted into temperature-specific advantages similar to the critic-free case. For hybrid RLHF approaches, TAMPO can similarly incorporate a combination of step-wise (critic-based) and trajectory-level (direct reward) signals to guide temperature adaptation.
>
> While conceptually straightforward, careful design is needed to mitigate potential bias or high variance from the critic’s estimates. We leave empirical evaluation in these settings to future work.

---

### Official Review · Reviewer_Eawz · 2025-10-31

**Soundness:** 2
**Presentation:** 3
**Contribution:** 3
**Rating:** 6
**Confidence:** 4

**Summary:**

This paper proposes Temperature Adaptive Meta Policy Optimization (TAMPO) that can dynamically adjust the temperature of large language models (LLMs) for reinforcement learning (RL)-based post-training of LLMs. More specifically, TAMPO consists of a hierarchical two-loop process: inner loop and outer loop. The inner loop optimizes the LLM policy, and the outer loop learns the temperature meta-policy. This paper evaluates TAMPO on five math reasoning benchmarks including AIME24, MATH-500, AMC23, Minerva, and OlympiaBench. The experiment results show that TAMPO can achieve higher scores than a base model (i.e., DeepSeek-R1-Distill-Qwen-1.5B) and GRPO-based post-trained models.

**Strengths:**

- S1. [Presentation] First of all, this paper is well written and organized.

- S2. [Novelty] The basic idea of learning LLM temperature for RL-based LLM post-training (i.e., meta-policy learning) seems novel.

**Weaknesses:**

- W1. [Performance] For meta-policy learning, TAMPO additionally calculates likelihoods at virtual temperatures when training the policy model. This increases the computational complexity of the GRPO-based LLM post-training. However, compared to the basic GRPO-based post-training (pass@1: 42.6%), TAMPO provides a slightly higher average accuracy (pass@1: 44.5%).

- W2. [Hyper-parameters] Even though this paper proposes to learn the temperature meta-policy, this may introduce additional hyper-parameters to search (e.g., EMA coefficient and sampling strategy).

**Questions:**

- Q1. [Overhead] Compared to the baseline method (e.g., GRPO-based post-training), how much computational complexity does the TAMPO have?

---

> ### Author Response · Authors · 2025-11-21
>
> We thank the reviewer for the thoughtful reading and constructive feedback. We are glad that the reviewer finds the paper well-written and recognizes the novelty of our approach. We address the raised concerns below.
>
> ---
> Q1: For meta-policy learning, TAMPO additionally calculates likelihoods at virtual temperatures when training the policy model. This increases the computational complexity of the GRPO-based LLM post-training. Compared to the baseline method (e.g., GRPO-based post-training), how much computational complexity does the TAMPO have?
>
> A1: One of the core design goals of TAMPO is efficiency. We achieve this by fully reusing inner-loop trajectories, thereby avoiding the need to generate any additional rollouts—rollout generation typically accounts for a dominant portion of GRPO training time.
>
> TAMPO additionally calculates likelihoods at virtual temperatures using Eq.(8) and Eq.(2) of the main manuscript, where the required token logits $z(o_{i,t} \mid s_{i,t})$ are already available from the inner-loop rollouts. The extra operations (Eq. (8) and (2)) are highly efficient and introduce negligible overhead.
>
> Empirically, TAMPO takes 178.42 seconds per training step, compared to 178.53 seconds for GRPO—an increase of only 0.6‰. This demonstrates that TAMPO’s meta-policy learning adds only a minimal computational cost while providing obvious performance improvements.
>
> ---
> Q2: Hyper-parameters
>
> A2: Thank you for the comments. TAMPO is designed to address the core challenge of dynamically selecting appropriate sampling temperatures during LLM RL—a task that is fundamentally difficult to hand-craft or tune manually at each training step. While TAMPO introduces a few fixed design hyperparameters (e.g., EMA coefficient, sampling strategy), these remain constant throughout training and are far easier to set than specifying a full temperature schedule.
>
> Dynamic temperature adaptation remains an important but underexplored aspect of LLM RL. We hope that TAMPO can serve as a practical step toward addressing this challenge and inspire further exploration in the community.

---

### Official Review · Reviewer_c2ti · 2025-11-02

**Soundness:** 3
**Presentation:** 2
**Contribution:** 2
**Rating:** 4
**Confidence:** 3

**Summary:**

The paper proposes Temperature Adaptive Meta Policy Optimization (TAMPO), a framework to dynamically adjust the sampling temperature of a LLM during reinforcement learning. Instead of treating the temperature as a fixed hyperparameter, TAMPO treats it as a learnable meta-policy that is optimized alongside the main policy. The approach operates in a hierarchical two-loop process: an inner loop updates the LLM’s policy (using a critic-free RL algorithm like GRPO) with trajectories sampled at a temperature chosen by the meta-policy, while an outer loop updates the meta-policy by rewarding temperatures that yield high-advantage trajectories. The authors evaluate TAMPO on five mathematical reasoning benchmarks (e.g., AIME, MATH, AMC, Minerva, Olympiad datasets), comparing against fixed and heuristically scheduled temperatures

**Strengths:**

1. The technical approach is sound and well-motivated. The paper identifies a clear limitation of existing LLM RL methods – the use of fixed or manually tuned exploration temperature – and offers a principled solution.
2. Significant practical strength of TAMPO is its decoupled outer-loop update mechanism, which enables online adaptation without additional rollouts。

**Weaknesses:**

1. Failure to Contextualize within the Meta-Gradient Literature: This omission is compounded by the paper's flawed positioning within the meta-learning literature it does cite.The paper does cite "meta-gradient methods" (Xu et al., 2018) for learning other hyperparameters like $\gamma$ and $\lambda$. It fails to cite the direct combination of these two concepts: "Meta-SAC: Auto-tune the entropy temperature of soft actor-critic via metagradient" (Wang & Ni, 2020), which applies a meta-gradient approach to this exact problem.
2. The paper asserts: "The trajectory likelihood is unimodal w.r.t. $T$". This unimodality is what justifies searching for a single likelihood-optimal temperature $T^{\star}$ (Equation 7). No direct evidence found in the manuscript. Figure 2 provides only 8 illustrative examples, which is not a proof.If this claim is false (e.g., if the likelihood function $\ell_T(\tau_i)$ is multi-modal), the entire basis for the update mechanism is flawed. The method would reinforce a local likelihood-optimal temperature, and the neighboring contributions (§3.3.2) would be meaningless. This is a critical gap in the paper's technical soundness.

**Questions:**

1. Which prior approaches to adaptive exploration or hyperparameter tuning in RL are most closely related, and how does TAMPO differ from them? The submission would benefit from discussing additional related works that were not cited. For example, methods that dynamically adjust exploration parameters (entropy, $\epsilon$-greedy schedules, or temperature annealing strategies in RL) and meta-learning approaches for RL hyperparameters deserve mention
2. What is the effect of TAMPO’s design decisions such as using a discrete candidate set of temperatures and the chosen range [0.6, 1.5]? The method currently relies on a fixed set $\mathcal{T}$ of 10 temperatures and updates a categorical distribution over them. This discretization might limit the precision of the optimal temperature found.
3. Can the authors provide more intuition or evidence about the underlying assumption that trajectory likelihood vs. temperature is unimodal?
4. The paper's outer-loop update contains two confusing choices.a) Why use sparsemax to normalize the likelihoods? Sparsemax is designed to create sparsity, which seems to contradict the stated goal of giving attenuated positive contributions to neighboring temperatures. Would a standard softmax not be more appropriate?b) Why use min-max normalization (Equation 12) for the final policy update? This is a simple heuristic.
5. Given the citation to meta-gradient methods, why not use a proper policy gradient update on the meta-policy $\pi(T)$, using the aggregated advantage as the advantage signal for the action $T_k$?

---

> ### Author Response · Authors · 2025-11-21
>
> We sincerely thank the reviewer for the detailed and constructive feedback. We are glad that the reviewer finds the technical approach well-motivated and recognizes the practical strength of TAMPO’s decoupled outer-loop update mechanism. We address the concerns and questions below.
>
> Q1: Fail to cite Meta-SAC.
>
> A1: We appreciate the reviewer’s suggestion on contextualization. Our submission does cite Meta-SAC (Wang & Ni, 2020) in the Related Work section; here we clarify the distinction.
>
> Meta-SAC optimizes an entropy-regularization coefficient (not the sampling temperature as ours even though it is referred to temperature coefficient) $\alpha$ via meta-gradients in RL, while our TAMPO performs a meta-policy update on the LLM sampling temperature by reusing inner-loop rollouts. Optimizing sampling temperature in LLM RL is challenging and under-explored.
>
> Challenges and Our Solution:
> Prior works such as Meta-SAC (Wang & Ni, 2020) and Xu et al. (2018) rely on fully differentiable pipelines to compute meta-gradients for hyperparameter optimization. In LLM RL, however, rollout generation and policy optimization are typically decoupled—rollouts are generated using lower-precision models for efficiency, preventing gradient backpropagation to the sampling temperature.
> Direct gradient-based optimization of the temperature meta-policy is therefore infeasible. Naive trial-and-error approaches (for obtaining rewards of different temperatures) are computationally prohibitive.
>
> TAMPO addresses this by reusing existing trajectories and evaluating their likelihood under virtual temperatures. This enables reward estimation for multiple candidate temperatures and meta-policy updates without additional rollouts.
>
> Conceptual Distinction:
> Meta-SAC adjusts the entropy coefficient $\alpha$ to trade-off exploration and exploitation in a traditional RL setting:
> $$
> J(\pi) := \sum_{t=0}^{T} \gamma^{t} \mathbb{E}{s_t, a_t \sim \rho\pi}
> \left[ r(s_t, a_t) + \alpha \mathcal{H}(\pi(\cdot \mid s_t)) \right].
> $$
>
>
> In contrast, TAMPO adapts the sampling temperature of an LLM, which rescales logits during token generation and directly modulates the exploration–exploitation trade-off in token sampling—a fundamentally different setting from entropy-regularized RL.
>
> ---
>
> Q2: No evidence found for “Unimodality” of trajectory likelihood.
>
> A2: The rigorous proof was included in Appendix B of our original submission; we have now highlighted it with bold in the revision to make it more visible.
>
> ---
>
> Q3: Discussing additional related works.
>
> A3: We thank the reviewer for the good suggestion and have added more related works in our revision. The meta-gradient methods as discussed in A1 are most closely related. The hyperparameters studied in (Xu et al., 2018; Wang & Ni, 2020) are differentiable, enabling direct optimization. In contrast, the sampling temperature in typical LLM RL frameworks is non-differentiable, rendering these methods infeasible (see A1 above). We propose the practical TAMPO approach, which efficiently adapts the sampling temperature by reusing existing rollouts without requiring gradient-based optimization.
>
> We added more citations and discussion. Traditional methods, such as $\epsilon$-greedy, temperature annealing, and upper confidence bounds (UCB), employ fixed or heuristic schedules to manage this balance (Sutton et al., 1998). Typically, $\epsilon$-greedy linearly or exponentially decays $\epsilon$ over time, temperature annealing gradually lowers the sampling temperature to reduce exploration, and UCB adaptively selects actions based on upper confidence bounds that balance estimated value and uncertainty. Sethi et al. (Sethi et al., 2025) view policy optimization as a continuous-time dynamical system and gradually decay the entropy regularization weight (typically as $1/t$).

---

> ### Author Response · Authors · 2025-11-21
>
> Q4: Effect of discrete temperature set.
>
> A4: Thank you for the question. Because the sampling temperature in LLM RL is non-differentiable, TAMPO is designed as a practical solution that avoids generating additional rollouts by reusing inner-loop trajectories. A discrete candidate set enables this: each rollout collected at a given sampling temperature can be evaluated under multiple virtual temperatures, greatly improving data efficiency for meta-policy updates.
>
> In practice, the trajectory likelihood varies smoothly and is unimodal with respect to temperature, so this discretization provides an accurate approximation of the optimal value. Empirically, expanding the grid to a finer interval (e.g., 50 temperatures) yields no additional performance benefit (see below table), indicating the current granularity is sufficient. The range [0.6, 1.5] is chosen based on common LLM RL practice: lower temperatures restrict exploration excessively, while higher temperatures lead to unreliable or low-quality generations.
>
> | Method | Avg Pass@1 | Avg Pass@8 | AIME24 Pass@1 | AIME24 Pass@8 | MATH-500 Pass@1 | MATH-500 Pass@8 | AMC23 Pass@1 | AMC23 Pass@8 | Minerva Pass@1 | Minerva Pass@8 | OlympiadBench Pass@1 | OlympiadBench Pass@8 |
> |--------|------------|------------|----------------|----------------|------------------|------------------|---------------|---------------|------------------|------------------|-------------------------|-------------------------|
> | TAMPO with 50 temperatures | 44.6 | 63.6 | 23.3 | 36.7 | 76.6 | 90.8 | 55.0 | 82.5 | 28.7 | 45.6 | 39.1 | 60.2 |
>
> ---
>
> Q5: Choice of sparsemax to normalize the likelihoods and min–max normalization for the final policy update.
>
> A5: We appreciate the reviewer’s insightful comment.
>
> Regarding the choice of Sparsemax normalization:
> We choose Sparsemax because it preserves the relative trends while producing sparse probability distributions—some relatively small outputs are mapped to zero. We found that directly applying softmax can cause the accumulated normalized values to become overly smooth.  Subsequently, we applied a scaling to softmax and observed that when the distribution is sharpened—for example, using a scaling factor of 0.3—it achieves performance comparable to Sparsemax and yields similarly strong results. Here is the result. Since using softmax requires introducing an additional scaling parameter to counteract its excessive smoothing behavior, whose inappropriate setting can easily hurt performance, we adopt Sparsemax in our implementation.
>
>
> | Method | Avg | AIME24 | MATH-500 | AMC23 | Minerva | OlympiadBench |
> |--------------------------------|------------|----------------|------------------|---------------|------------------|-------------------------|
> | Softmax (scaled, τ = 0.3, tuned best)      | 44.7   | 26.7           | 74.8             | 55.0          | 26.8             | 40.2                    |
> | Sparsemax (no scaling needed)  | 44.5   | 23.3           | 76.8             | 55.0          | 27.9             | 39.6
>
> Regarding the choice of min–max normalization:
> We use this simple heuristic to map temperature-specific advantages into the [0, 1] range, establishing a consistent scale across different training steps. This makes the accumulated historical scores comparable and further facilitates temperature sampling. Other transformation functions could also be used.
>
> ---
>
> Q6: Given the citation to meta-gradient methods, why not use a proper policy gradient update on the meta-policy?
>
> A6: Unlike Meta-SAC (Wang & Ni, 2020) and Xu et al. (2018), where the parameters to be optimized are differentiable, rollout generation and policy optimization in LLM RL are typically decoupled—given a sampling temperature, rollouts are generated using lower-precision models for efficiency—preventing gradient backpropagation to the sampling temperature.
> Direct gradient-based optimization of the temperature meta-policy is therefore infeasible.
>
> As an alternative, a naive trial-and-error strategy is to generate rollouts for different temperatures to obtain their rewards for meta-policy optimization. But this is computationally prohibitive and suffers from low sampling efficiency. We propose TAMPO, which works with the existing decoupled pipeline and requires no extra rollouts for effective and efficient meta-policy optimization.

---

### Author Response · Authors · 2025-12-01

We sincerely thank all the reviewers for their constructive feedback. We are encouraged that reviewers recognize:

**Novelty & Motivation** – Recasting sampling temperature as a learnable meta-policy is well-motivated (Reviewer c2ti), novel/fresh (Reviewer Eawz, aLbY), and interesting (Reviewer a4qg).

**Technical Soundness & Efficiency** – TAMPO’s outer-loop meta-policy is practical and efficiently reuses rollouts without additional sampling cost (Reviewer c2ti, a4qg, aLbY).

**Consistent Empirical Gains** – TAMPO demonstrates consistent improvements across multiple reasoning benchmarks” (Reviewer a4qg).

**Clear Presentation** – The paper is well-written and clearly structured (Eawz, aLbY).

Regarding the concerns and suggestions raised by each reviewer, we have addressed them as thoroughly as possible and provided detailed, point-by-point responses. Below, we summarize our responses to the core concerns:

**Reviewer c2ti (score: 4, confident 3)**: 1) We clarified that prior meta-gradient methods (Xu et al., 2018; Meta-SAC) assume differentiability, whereas LLM RL rollouts are generated via non-differentiable sampling and low-precision models, making direct meta-gradients infeasible. TAMPO’s virtual-likelihood–based update is specifically designed for this LLM RL setting.  2) We highlighted the formal proof on unimodality claim already provided in Appendix B in our original submission and improved visibility. 3) We additionally incorporated ablation studies for the design choices, e.g., the discrete temperature granularity, softmax alternative.

We believe the key concerns as indicated in weakness were addressed (one is about citation, the other is the unimodal proof which was already in the Appendix B).

**Reviewer Eawz (score: 6, confident 4)**: For the concern on increased computational complexity, TAMPO adds negligible compute (~0.6‰), since virtual temperature likelihoods reuse existing logits. The high efficiency was recognized by Reviewer c2ti, a4qg, aLbY.  Regarding hyperparameters, TAMPO replaces a hard-to-design full temperature schedule with a few simple, fixed hyperparameters (e.g., EMA), where no hyperparameter tuning across steps is required.

**Reviewer a4qg (score: 4, confident 3)**: We added further experiments using a different base model and benchmark that demonstrated the strong generalization capability of TAMPO beyond mathematical reasoning. We clarified the optimal solution of the iterative optimization, elaborated on its theoretical foundation, and discussed the broader applicability of our method.

**Reviewer aLbY (score: 6, confident 4)**: We added experiments using another base model and benchmark to demonstrate TAMPO’s generalization ability. We reported the mean, standard deviation in the rebuttal, which shows consistency with one-run results. We added additional ablation studies on the discrete temperature set to examine the influence of finer granularity. We further discussed potential future extensions and improvement directions.

**Manuscript Revisions** – We have accordingly revised the manuscript to clarify related work, highlight proofs, enrich experiments, and refine several descriptions. All changes are marked in blue in the revised version.

Should any new or remaining issues arise, we would be glad to provide further clarification.

---

### Meta-Review · Area_Chair_CGq7 · 2026-01-05

**Summary:**

Here is a summary of the reviewers' concerns:
- Limited empirical justification: the experimental results were reported for one model(GRPO) with one run and one type of tasks which limited the justification for reinforce learning requiring at least 3-5 independent runs, diverse models(critic-based or hybrid RLHF methods)/tasks to demonstrate the effectiveness of the methods.
- Unclear generality: All experiments are conducted on mathematical reasoning tasks, one Model. It is not clear how to generalize to other base models and task domains.
- Limited temperature search space:  The range (0.6-1.5) and interval (0.1) of the candidate temperature set are manually preset without an adaptive adjustment mechanism or continuous temperature optimization for different models or task scenarios.
- Unclear scalability: All experiments are conducted on a 1.5B parameter model. It is unclear how well the method generalizes to larger models.
- Additional computation overhead: For meta-policy learning, TAMPO additionally calculates likelihoods at virtual temperatures when training the policy model. This increases the computational complexity of the GRPO-based LLM post-training.
- Missing citation and discussion of related work on the Meta-Gradient Literature.
- Additional hyper-parameters: The proposed algorithm introduces additional hyper-parameters to search (e.g., EMA coefficient and sampling strategy).
- Lack ablation study for different base models.
- Lack deep analysis of how adaptive temperature affecting reasoning quality.
- Inefficiency in multi-task settings or under highly heterogeneous prompts due to shared temperature distribution across the entire dataset.
- Policy bias due to strongly coupled policy updates, reducing the stability of temperature adaptation.
- Typo: Line 356 and 357, “using initial learning rate 1x10e-6” but “minimum learning rate of 0.1” is confusing and counterfactual.
- Lack theoretical/intuitive perspective of  iterative optimization.
- Lack theoretical justification of the unimodal assumption of the  trajectory likelihood.

**Reviewer Concerns:**

Here are the reviewer concerns addressed by the rebuttal:
- Limited empirical justification: added the experimental results with 3 runs on 5 benchmarks.
- Additional computation overhead: The extra operations (Eq. (8) and (2)) are highly efficient and introduce negligible overhead.
- Generalization to other base models/domains, Scalability: conducted additional experiments to evaluate TAMPO’s generality across different base models and domains. Discussed the applicability to critic-based or hybrid RLHF methods.
- Typo: corrected it.
- Policy bias: TAMPO is robust due to Batch and temporal aggregation, smooth temperature–likelihood relationship,
- theoretical/intuitive perspective of  iterative optimization
- analysis of how adaptive temperature affecting reasoning quality
- Lack theoretical justification of the unimodal assumption: The rigorous proof was included in Appendix B.
- Missing citation and discussion of related work: added more citations and discussion.

The concerns that are partially addressed:
- Lack explicit conditioning on the prompt, subtask, or training stage
- Limited temperature search space
- Additional hyper-parameters: While TAMPO introduces a few fixed design hyperparameters (e.g., EMA coefficient, sampling strategy), these remain constant throughout training and are far easier to set than specifying a full temperature schedule.

**Reviewer Scores:**

Reviewer c2ti raised 3 concerns, most of the concerns were addressed by the rebuttal. Reviewer c2ti might increase the score to 6.

Reviewer Eawz raised 2 concerns, most of his concerns are addressed fully or partially by the rebuttal. Reviewer Eawz might keep the positive score.

Reviewer a4qg raised 5 concerns, most of the concerns were addressed by the rebuttal. Reviewer a4qg might increase the score to 6.

Reviewer aLbY raised 6 concerns, the authors addressed 4 of them fully and the rest 2 partially. Reviewer aLbY might keep the positive score.

---

### Decision · Program_Chairs · 2026-01-26

Accept (Poster)